# Leveraging Computational Intelligence Techniques for Defensive Deception: A Review, Recent Advances, Open Problems and Future Directions

**DOI:** 10.3390/s22062194

**Published:** 2022-03-11

**Authors:** Pilla Vaishno Mohan, Shriniket Dixit, Amogh Gyaneshwar, Utkarsh Chadha, Kathiravan Srinivasan, Jung Taek Seo

**Affiliations:** 1School of Computer Science and Engineering, Vellore Institute of Technology (VIT), Vellore 632014, India; pillavaishno.mohan2019@vitstudent.ac.in (P.V.M.); shriniket.dixit2019@vitstudent.ac.in (S.D.); amogh.gyaneshwar2019@vitstudent.ac.in (A.G.); kathiravan.srinivasan@vit.ac.in (K.S.); 2School of Mechanical Engineering, Vellore Institute of Technology (VIT), Vellore 632014, India; utkarsh.chadha2018@vitstudent.ac.in; 3Department of Computer Engineering, Gachon University, Seongnam 13120, Korea

**Keywords:** defensive deception, machine-learning, deep learning, computational intelligence, honeypots, moving target defense

## Abstract

With information systems worldwide being attacked daily, analogies from traditional warfare are apt, and deception tactics have historically proven effective as both a strategy and a technique for Defense. Defensive Deception includes thinking like an attacker and determining the best strategy to counter common attack strategies. Defensive Deception tactics are beneficial at introducing uncertainty for adversaries, increasing their learning costs, and, as a result, lowering the likelihood of successful attacks. In cybersecurity, honeypots and honeytokens and camouflaging and moving target defense commonly employ Defensive Deception tactics. For a variety of purposes, deceptive and anti-deceptive technologies have been created. However, there is a critical need for a broad, comprehensive and quantitative framework that can help us deploy advanced deception technologies. Computational intelligence provides an appropriate set of tools for creating advanced deception frameworks. Computational intelligence comprises two significant families of artificial intelligence technologies: deep learning and machine learning. These strategies can be used in various situations in Defensive Deception technologies. This survey focuses on Defensive Deception tactics deployed using the help of deep learning and machine learning algorithms. Prior work has yielded insights, lessons, and limitations presented in this study. It culminates with a discussion about future directions, which helps address the important gaps in present Defensive Deception research.

## 1. Introduction

Advanced cyber defenses must provide a quick response against attacker activities in real-time scenarios. They demand clever defense systems that can automatically react to adversarial conduct and evolve with time as the progress of the attack. Before running a defensive action, the AI method utilized by the defensive system should be able to have the foresight and analyze the pattern of an attacker to take appropriate defensive measures. Adaptive or active cyber security, in which a system plans and uses defense techniques automatically in response to an identified suspicious activity without human intervention, is growing rapidly, but it has not yet been extensively adopted.

Cyber Deception is one of the major techniques in cyber defense research. In comparison to standard security safeguards, deception-based systems operate fundamentally differently [1,2,3,4,5,6]. Traditional security measures are employed in response to the actions of an attacker, detecting or preventing them, whereas deception-based measures are used in anticipation of such actions, manipulating attackers’ perceptions and thus inducing adversaries to take decisions that are advantageous to systems which the adversary is targeting. 

Deception is especially significant in military-style attacks that are time sensitive, such as those carried out by cyber terrorists, where simply postponing the attack with the help of deceptions could be crucial until a permanent defense is developed [7]. Both insider and outsider attacks can be prevented using Deception. These days machine learning has emerged as an effective technology that provides us with a wide range of applications ranging from recognition of patterns, image identification, image, and video processing, making predictions, virus or malware detection, autonomous driving, and other application scenarios [8,9,10,11,12,13,14,15,16,17,18,19,20,21,22,23,24,25,26,27,28,29,30,31,32,33,34,35,36,37,38,39]. The advantages of machine learning algorithms can be extended for deploying Defensive Deception frameworks [40,41,42,43,44,45,46,47,48,49,50,51,52,53,54,55,56,57,58,59,60,61,62,63,64,65,66,67,68,69,70,71,72,73,74,75,76,77,78,79,80,81,82,83,84,85,86,87,88,89]. Deception has been employed in honeypots, which are legal traps and honeynets (honeypot networks), as a defensive tool for information systems to keep attackers occupied [90,91,92,93,94,95,96,97,98,99,100,101,102,103,104,105,106,107,108,109,110,111,112,113,114,115,116,117,118,119,120,121,122,123]. Honeypots are systems that exist solely to promote attacks in order to collect data. Interconnected honeypot networks are known as honeynets. Some honeypots employ deceptions such as phony files to entice attackers to stay away from actual resources for a while. Moving target defense, a type of deception technology, makes an attacker’s work more difficult by adding unpredictability to the attack area and changing information quickly. By incorporating falsehoods and obscuring real facts, Deception can add a new level of ambiguity. It can immensely affect the decision-making of an attacker, forcing them to squander time and effort.

Furthermore, a defense can utilize cyber Deception to give the attacker the wrong impression. This erroneous notion can generate ripple effects throughout the cyber death chain, disrupting several attacks over time. There are two major promising paths for developing Defensive Deception tactics in this literature. First, attacker and defender strategies have been commonly described using machine learning, with the defender employing Defensive Deception strategies to confuse or mislead attackers into choosing suboptimal or inferior strategies. Second, this article discusses deep learning-based Defensive Deception approaches implemented in recent cyber security advancements. The article then progresses with various taxonomies used in Deception and their description. Finally, the article concludes with future research directions and solutions for the same.

### 1.1. Contribution of this Survey

Our contribution can be summarized as follows:This is the first survey that briefly discusses the application of various Machine learning and deep learning methods in the implementation of Defensive Deception and its technologies.Discussion on new techniques in Defensive Deception such as Genetic Algorithms, Multi (Intelligent) Agents, DBN, SOM, etc., along with the traditional Computational Intelligence techniques such as KNN, Random Forest, ANN, DNN, etc.Detailed tabular summary of works on Machine Learning and Deep Learning Techniques in Defensive Deception are included. The summary provides the model, key contributions, and limitations for the same.A brief description of various methods to implement Defensive Deception has been provided. This includes Perturbation, Moving Target Defense, Obfuscation, Mixing, Honey-x, and Attacker Engagement.Classification of several deception categories and commonly used datasets have been mentioned.Finally, the paper describes various open challenges present in Defensive Deception and future research directions for further improvements in this field.

Table 1 presents the current review articles of the CI-enabled techniques in defensive deception.

### 1.2. Survey Methodology

#### 1.2.1. Search Strategy and Literature Sources

Databases such as ACM Digital, IEEE, Science Direct, etc., were used to find relevant articles. The keywords utilized were: Defense Deception, Fraud Detection, Cyber Defensive Systems, etc., alongside some other keywords relating to the possible fraud types. A total of 1138 non-duplicate articles were found from these databases initially.

#### 1.2.2. Inclusion Criteria

The articles included were based on their relevance. The articles were included based on the novelty of this review’s topic and appropriate language, and only English articles were included.

#### 1.2.3. Elimination Criteria

The eliminations of the articles are based on abstract screening, then based on full text and data extraction in the next iteration. The articles were eliminated due to lack of relevance, duplicate articles, articles not in the English language, or poorly written manuscripts.

#### 1.2.4. Results

There were 1138 articles shortlisted from various databases, and after inclusion/exclusion criteria, 77 articles were included for the review, which kept direct relevance with the defense deception; Figure 1 shows the PRISMA implementation for the same.

### 1.3. Survey Structure

This survey is prepared by referencing more than 70 research articles. Section 1 of this article consists of a brief overview. The selection process involved for the referenced articles is discussed, and a brief comparison has been performed for the various surveys involved. Section 2 discusses various CI-enabled techniques applied in Defensive Deception technologies. This section is divided into two major subsections. The first subsection includes a brief description of various machine learning algorithms applied in Defensive Deception technologies. The second subsection consists of deep learning algorithms and various applications to implement Defensive Deception technologies. In Section 3, we have described frequently used datasets in our survey, the various Defensive Deception taxonomies used and their implementation in real-world Defensive Deception technologies. Section 4 includes various open problems present in Defensive Deception and a brief description of future research directions. Finally, Section 5 includes the conclusion of this article, followed by the list of references at the end.

## 2. CI-Enabled Techniques Used in Defensive Deception

Computational intelligence consists of two substantial branches of artificial intelligence technologies: deep learning and machine learning. These methods can have a wide range of applications in Defensive Deception technologies. By merging autonomic computing and cyber Deception, we can obtain an early defender advantage and counter attacker behaviors through automatic adaptation. Article [8] proposes implementing the adaptive deception framework, which involves a tiny network consisting of two Windows 7 client computers and a database server. One hundred runs were performed for four different scenarios where the attacker tried to access this network. For the first control condition, no obstacles were present. As a result, all 100 runs were a success for the attacker, with an average run time of 250.05 s. For static decoys condition, decoys are pre-configured and pre-deployed. The attacker only succeeded 42 times in this situation, was unable to exploit and pivot 19 times, and failed to exfiltrate the database 39 times. The average time to success was 261.80 s, which was somewhat higher than the control average. In the delay condition, decoys are a pre-deployed but adaptive deception system with delay. In this condition, the attacker was successful 40 times, had 27 exfiltration failures and 23 pivot failures. The average successful run took 630.23 s. In the deny condition, decoys are a pre-deployed and adaptive deception system with denying. The attacker was successful 11 times, had 78 pivot failures and 11 exfiltrating failures. The average successful run time was approx. 256.64 s [8]. This article showed how the autonomous deception framework increased the attacker runtime by 175% and reduced the successful runs by 89%, resulting in an optimal defense strategy.

### 2.1. The Evolution and Overview of AI-Enabled Techniques

In the early stages of AI technology, we majorly tackled cyberspace threats using machine learning (ML) techniques. Although machine learning is extremely strong, it relies majorly on feature extraction. Researchers began studying deep neural networks, often referred to as Deep learning, a sub-domain of machine learning, in response to glaring problems in classical ML. Traditional ML and DL vary in that DL methods can be used directly for training and testing the original data without having to Remove or change their characteristics [1]. In the last few years, DL algorithms have shown a performance improvement of about 20–30% in image processing, natural language processing, and text recognition, and had a significant impact on the development of AI and have a major application in Defensive Deception technologies [1].

### 2.2. Machine Learning Techniques

ML algorithms are majorly used in AI systems to extract models using raw data. Finding ML solutions includes four major steps.

Extracting the features.Selecting an appropriate machine learning algorithm.After evaluating different algorithms and adjusting parameters, training the models, and selecting the model with the best performance.Making predictions for the unknown data with the help of the trained model [1].

The most frequent supervised approaches are those based on supervised machine learning algorithms, which collect large datasets and classify an account as either person or bot. Machine learning includes several strategies that can improve the accuracy of protection. When used effectively, ML powerful algorithms create a learning environment for systems, accomplishing tasks such as spotting known/unknown malicious attacks [2,3,6]. Figure 2 lists all the ML techniques utilized in Defense Deception.

#### 2.2.1. Naïve Bayes

The Bayes conditional probability rule is used in Naïve Bayes (NB), a classification tool. Every attribute along with the class label is treated as a random variable, then the naive Bayes algorithm selects a class for the newly fetched observation that maximizes its probability following the values of the various attributes, provided that the attributes are independent [1,16]. Although Naïve Bayes classifiers weaken when the features are derived from dependent events, they are extensively used because they assume a naive assumption (that every feature is derived from independent events) and can still produce acceptable results [5]. Naïve Bayes analysis works well for deception planners, taking the suitability of Deception into account and planning the type of Deception that needs to be deployed [11,30,113,120]. Naïve Bayes classifiers are widely used for email spam detection and network intrusion detection, which involves deceiving in order to cause harm to the system [12]. The probabilities of the three hypotheses, “network is down,” “bugs in the system,” and “deception,” can be calculated when a download attempt has been made as well as when an attempted modification has occurred, using a Naive Bayes approach. As a result, despite its low initial likelihood, the contradictory signs make Deception more feasible than the other hypotheses [23]. Naive Bayes is a useful categorization method that is simple to understand, and it is especially useful when the inputs include many dimensions [24].

#### 2.2.2. Decision Tree

The decision tree method is majorly used for extracting a set of inferences by analyzing the derived rules from a couple of training datasets or samples. The decision tree first finds a feature that can categorize the data samples iteratively. After each division, rules are generated for each part of the category. It, in turn, results in a tree-like structure. The process continues until only one class is identified for the data samples [1,5,16]. Because it reveals the result of choice based on feature values, the methodology can be extensively used for detecting cybersecurity issues. This can be achieved by classifying the observed cybersecurity events or occurrences as either being legitimate or an attack.

Furthermore, we can classify data in real time once the tree is defined [5]. We can deceive adversaries by employing probabilistic decision trees to make decisions. These trees can be built using grammar which specifies how a system should react in case of security threats. This technique can be built with the help of a historical dataset (playback) and a network simulation in real time [13]. Machine-learning-based techniques such as the ID3, CART and C4.5 can be used to grow these trees. Leaves indicate predictions, while branches represent feature combinations. Credit cards, auto insurance fraud and corporate fraud involve decision trees. The classification and regression trees, also known as the CART technique, are prominently used to detect and predict the impact of false financial statements [24]. When we have a group of honeypots (a honeynet), rather than just one, a decision tree is more useful to decide which honeypot configuration is best to deploy according to the given scenario. We can also independently test other techniques to determine how well they work and what risks they entail. This can be achieved by calculating the average benefit for several honeypots and honeynet layouts, and the one with the highest average benefit can be chosen [62,63,121].

#### 2.2.3. k-Nearest Neighbour

The k-Nearest Neighbour, commonly known as the k-NN approach, learns with the help of data samples to build classes or clusters. The proposal for k-NN was made as a non-parametric form of pattern analysis [73] which can be used for determining the fraction of data samples in a neighborhood that can produce a consistent probability estimate. To form clusters, the neighborhood is first established with the help of a k-number of data samples, usually based on a distance measure (Euclidian distance, Manhattan distance, etc.). When a dataset sample is newly introduced, it is grouped with one of the clusters based on the votes of all k neighbors. Even for tiny values of k, this strategy is computationally challenging. However, it is appealing for intrusion-detection systems to learn from new traffic patterns and detect zero-day attacks, which are attacks that are not yet known to the vendor or general public [5]. After the attack has been detected correctly, we can deploy appropriate deception decoys to protect the resources.

#### 2.2.4. Random Forest

Random forest works by creating various decision trees from an arbitrarily selected subset of training samples and variables. A random forest classifier is simple to learn and use and quick to test. This learning method is well known for handling nonlinearity and outliers and compatibility with big datasets simultaneous training. A strategy based on decreasing entropy once a dataset is split into separate qualities is known as the information gain feature [3]. A list of 13,884 SQL statements was utilized in the dataset, compiled from multiple sources. 12,881 are malicious (SQL Injections), while 1003 are legitimate. They removed extreme values and outliers during data pre-processing. When 10-fold cross-validation is applied to the dataset, it has an accuracy of 99.1% for SQLI prediction [27]. They used Random Forest to classify the material polluters and then used conventional boosting and bagging and alternative feature group combinations to improve the findings. The authors were able to obtain a higher rate of social adversary collection with the help of a random forest model and, as a result, were able to improve the social honeypots. The upgraded Honeypot collected social enemies 26 times faster than an unaltered social honeypot [25] based on a random forest classifier evaluation.

#### 2.2.5. Support Vector Machine

In order to perform machine learning tasks, Support Vector Machine—commonly known as SVM learning—is a prominent and widely used method. Support vector machine falls under supervised machine learning technologies for categorizing data. This division methodology employs a series of training examples, each of which is classified into one of two groups. After that, the SVM is used to create a model that can predict if a new sample instance belongs to one of two categories using a separating plane. This categorization method aids systems in providing tiny sample sets with improved learning capabilities. The SVM approach can be widely applied in network intrusion detection, online page identification, and facial recognition applications. When used in intrusion detection systems, SVM offers benefits which include high training and decision rates, insensitivity to input data dimension, and constant correction of multiple parameters with a boost in training data, enhancing the system’s ability to self-learn [1,3,5,16,18,122,123]. Email spam detection is a successful implementation performed using SVMs [12]. Support Vector Machines can outperform neural network models and cluster and classify outliers using a higher dimensional feature space obtained from the training dataset [40,41]. To find malicious profiles and obtain data from these profiles, the authors used feature-based techniques and honeypot strategies and then evaluated the data using Support Vector Machines (SVM) and other machine learning algorithms [42]. They coined the term “active honeypots,” which are Twitter accounts that can catch as many as ten new spammers in a single day. They used Twitter to find 1814 accounts and looked at the essential characteristics of active honeypots. Furthermore, the authors investigated the impact of unbalanced datasets on detection accuracy for various ML methods using a suite of ML techniques, including SVM [28,39,43].

#### 2.2.6. Ensemble Models

Ensemble approaches are useful for security use during the testing or inferring phase. A vast body of work aimed at designing Moving Target Defense systems, commonly known as MTDs, highlights the security benefits while ignoring the performance drawbacks. It is worth noting that the performance impact of MTDs might occur for various reasons. Each MTD ensemble system configuration has an efficiency cost attached, and switching to a high-cost arrangement influences performance [30]. Ensemble models can demonstrate the MTD security benefits by contrasting them with an unaltered system configuration [54,55,66,86,123].

#### 2.2.7. Genetic Algorithms

Even though game-theoretic MTD approaches are the most popular, other techniques such as genetic algorithm is another viable option. Genetic algorithms (GAs) are frequently employed to maximize solutions’ optimality. Furthermore, in some fully dispersed setups, ensuring a centralized organization to make MTD decisions based on GAs could be impossible [31].

#### 2.2.8. Multi (Intelligent) Agents

These agents provide proactive cyber-defense techniques such as gathering data, assessing security, monitoring network state, attack detection and countermeasures, malefactor deception, etc. Machine learning techniques applied to the usual interaction between agents in a multi-agent system, for example, can result in coordinated actions and plans emerging on their own Multi-agent system (MAS). Agents are expected to gather information from various sources, use partial knowledge, predict the intentions and behaviors associated with other agents, make decisions according to the actions of other agents and attempt to deceive opposing team agents [9,10].

Table 2 provides an executive summary of the machine learning research works in Defensive Deception.

### 2.3. Deep Learning Models

Figure 3 shows all the current deep learning models in defensive deception, which will be explained in this section below.

#### 2.3.1. Artificial Neural Network

Artificial Neural Networks (ANN) have computational abilities that help them simulate functional and structural aspects of neurons present in biological systems. They are capable of performing parrel processing of information and high-speed decision-making. These properties make them suitable for attack pattern recognition, classification, and response selection. These can not only be used for IDPS (Intrusion detection and prevention system), but there are also proposals for their application in DOS, malware, worm, and spam detection systems along with forensic investigations [4]. An ANN application was employed in a cybersecurity investigation that used the Cascade Correlation Neural Network (CCNN), which adds additional hidden units to the currently present hidden layers under the algorithm. In this study, the CCNN allows the network to analyze and learn from traffic patterns generated by desktop platform to detect port scanning of mobile networks without requiring the entire network to be retrained with the original data [5]. Another advantage of ANN is that it can detect zero-day attacks due to its ability to learn from previous instances. For example, labeled training data, including traffic patterns generated from DoS attack instances, were fed into ANNs, after which the neurons were able to detect hidden DoS attacks [5]. Users can utilize ANN in cloud-based models to get useful class probability information while reducing the chances of an adversary stealing the model. The last activation layer of the model can be perturbed using ANN, slightly modifying the output probabilities. The adversary is forced to ignore the class probabilities, making it necessary to use more queries before successfully performing an attack. The evaluation demonstrates that such a defense can reduce the stolen model’s accuracy by at least 20%, or 64 times increase in the number of queries necessary for an adversary, all with a small impact on the protected model’s accuracy [61].

#### 2.3.2. Recurrent Neural Networks

Unlike typical feed-forward neural networks, Recurrent neural networks use directional loops to process sequence data and manage contextual correlation among inputs [1]. RNN (Recurrent Neural Networks) can handle time-series data and raw input feature values and capture data involving changes over time. Within five seconds of running the report, a collection of RNNs can assess whether traffic is malicious or benign with a 97 percent accuracy rate [3]. Using the KDD CUP 1999 dataset, they used Recurrent Neural Networks (RNNs) to identify intrusions and achieved a full detection rate with only a 2.3 percent false alarm rate [46]. The deception jammers were integrated into legitimate systems to make them harder to recognize and more desirable targets. To improve the fidelity of the additional decoy devices to the actual system, three properties are maintained: a protocol, parameters, and logic of the deployed false devices. The authors used a dataset collected over a year to train a recurrent neural network (RNN) to understand such system properties. RNN (Recurrent Neural Networks) was also utilized to generate fake devices that looked real, based on a year of observations of device behavior in a CPS [25].

#### 2.3.3. Deep Neural Network

DNN (Deep Neural Networks) consists of neural networks with a big number of disguised layers [15]. DNNs are a subset of ANNs. Multiple hidden layers are employed in DNNs, allowing various algorithms to analyze variables that would otherwise go unnoticed if only a single layer were used [5]. DNN outperforms neural networks in terms of capacity to fit complex mappings. DNN collects features layer by layer, combining low-level and high-level features in the process. Deep Belief Networks, Stacked Autoencoder and Deep Convolution Neural Networks (DCNN) are three regularly utilized DNN models [32,51,52]. The paper’s authors introduced the MTD framework for DNNs, which improves their security and resilience against adversarial assaults. The ideal switching strategy for MT Deep is the Stackelberg equilibrium of the game, which reduces misclassification while retaining excellent classification accuracy for genuine system users on neutrally produced images [31,53].

#### 2.3.4. Deep Belief Network

It is a probability generation model which uses several limited Boltzmann layers to generate probabilities. DNN collects features layer by layer, combining both high-level and low-level features in the process. DBN, SAE, and DCNN are the three most often utilized DNN models. It is a probabilistic unsupervised deep learning algorithm. DBNs comprise layers of Restricted Boltzmann Machines [32,52], followed by a feed-forward network for the fine-tuning step [32,52]. Combining a Deep Belief Network and a probabilistic neural network can result in a novel intrusion detection system that can be used to configure deception measures. The original data were turned into low-dimensional data in this method, then DBN (a nonlinear learning algorithm) identified the main properties from the original data. A particle swarm optimization technique optimized the number of hidden-layer nodes per layer. The low-dimensional data were then classified using a PNN (probabilistic neural network). The “KDD CUP 1999” dataset revealed that this technique outperforms classic PNN, PCA-PNN, and original DBN-PNN without simplification [1]. After the intrusions are detected, appropriate deception decoys can be deployed for protection.

#### 2.3.5. Deep Reinforcement Learning

Deep Reinforcement Learning is a hybrid of reinforcement and deep learning. Reinforcement learning is a branch of machine learning that involves executing appropriate action to maximize the reward in a given situation. It is used to determine the best possible conduct or path to pursue in a given situation [32,56]. Deep Reinforcement Learning (DRL) estimates difficult functions with high-dimensional inputs using a neural network. The addition of deep learning to traditional RL approaches improves the ability to capture the huge scale of numerous Internet-connected systems, such as mobile networks and IoT devices [15]. DRL could develop a low-dimensional version of high-dimensional data, which is quite compact in nature. DNN is a powerful complement, allowing it to be used for the cyber security of vast networked systems [15]. There has been a recent surge of research on using RL to select an adaptive configuration strategy to maximize the impact of MTD, with particular emphasis on the dynamic environment, reduced resource consumption, usability, partially observable environments, and multi-agent scenarios that include both the system’s characteristics and the adversary’s observed activities. Through a compromise between usability and security, we can investigate the potential of RL in reconfiguring defenses [15]. Creating hybrid Defensive Deception tactics that incorporate machine learning and game theory: Other protection mechanisms have been considered using machine learning-based game-theoretic techniques. Other researchers have developed hybrid approaches that combine reinforcement learning and game theory, using RL as one of the most important parts of the machine learning technique. As in other attack–defense games, Reinforcement Learning’s reward functions can be utilized to create players’ utility functions and allow an RL agent to determine an ideal strategy [25,57,58].

#### 2.3.6. Self Organizing Maps

A self-organizing map (SOM) is a neural network that uses unsupervised learning instead of supervised learning [16]. Self-organizing maps (SOMs) are a popular data visualization tool enabling the representation of a multi-dimensional dataset on a two-dimensional or three-dimensional map. In other words, they help us to perform dimensionality reduction [5,17]. The approach learns by searching for correlations in data samples. Adjacent data samples have more in common with each other than samples further apart, resulting in data clustering and a map as an output. Because SOMs are computationally intensive, they are unsuitable for real-time systems. Their main advantage is their capacity to visualize data, which is important for detecting network irregularities and understanding the best deception decoy deployed [5]. Table 3 summarizes works on Deep Learning Models in Defensive Deception.

## 3. Defensive Deception

### 3.1. Datasets Used in Defensive Deception

Table 4 below lists the various defense deception datasets utilized in various research works. Figure 4 portrays the various methods to implement Defensive Deception.

### 3.2. Perturbation

Perturbation is a technique for limiting the leakage of sensitive data by inserting noise [20]. A defender can use perturbations to initiate Defensive Deception via external noises [19]. The method used in this article allows clients to access critical data summary facts that are not altered and do not compromise data security. Perturbation could be used to build a detailed counterplan that balances disruption with the potential to deceive the attacker based on the believability of the ploys deployed [36]. Table 5 discusses the classification of several deception categories.

### 3.3. Moving Target Defense

Moving target defense can also be used to build an RL-CRM (Reinforcement learning—Cyber-resilient mechanisms) that attracts jammers to attack a bogus route to safeguard actual communication [15]. MTD is related to Defensive Deception in that it aims to enhance attackers’ confusion or ambiguity, preventing them from escalating or failing their attacks to the next level. The major difference is that MTD does not actively mislead attackers with misleading information, whereas Defensive Deception frequently entails using fake items or details to cause aggressors to generate false ideas and be tricked into making inefficient or weak attack judgments. MTD’s major trait is that it focuses on modifying system configurations with greater understanding and efficiency, whereas Defensive Deception focuses on changing the attacker’s perspective [25]. An MTD’s purpose in altering the Prevention Surface is to keep the attacker unsure of the defense mechanism in place, forcing the adversary to invest more resources and devise more complicated techniques to unencrypt the data. SDN (Software Defined Networking) architecture may be vertically divided into three tiers: data, control and application plane. When utilizing a Moving Target Defense that switches among several configurations, one would like to think that it improves the integrity of the implemented system while having no detrimental influence on legitimate users’ efficiency. Quantitative analysis based on usability and security metrics has been used for the same. During the creation, installation, and assessment of Moving Target Defenses, their classification had aided in finding several areas that had been underexplored (MTDs). Although difficult to implement, the mobility of multiple platforms within a single framework can provide more security benefits than a single platform movement. In order to research in this field, one must first discover sets of setups that are consistent (in terms of performance) across multiple surfaces. They attempted to classify a variety of existing works using this nomenclature. E.g., in a project, a MTD is a defense used for the mobility of detecting surfaces with fixed cycle flipping framed as a multi-stage game that simulates basic use cases and assesses the security and performance of various defenses in these situations. They also discovered that hierarchical techniques such as Software Defined Networking aid in implementing MTD remedies with little networking performance effect [30]. Moving target defense can also be used to build an RL-CRM (Reinforcement learning—Cyber-resilient mechanisms) that attracts jammers to attack a bogus route to safeguard actual communication [15].

### 3.4. Obfuscation

Obfuscation defenses divert an enemy’s resources by displaying and diverting them to decoy targets rather than the network’s genuine resources and providing fake data mixed with real (i.e., valuable) data [20]. The main goal of obfuscation is to slow down the attacker’s movement within the network and systems [21]. A leader–follower game (also known as the Stackelberg game) was modeled between an obfuscation technique designer and a possible attacker. To counter optimum inference attacks, the authors devised adaptive techniques. They anticipated that when consumers share sensitive data with untrustworthy entities, they will take precautions to secure it. This allows users to disguise data before sharing it by adding noises. The attacker has access to sensitive user information and obfuscation-related noises [59]. Data obfuscation has several advantages over other DD approaches, including honey-x techniques, which are designed to deceive enemies into making suboptimal or weak attack decisions by providing incorrect information, ease of deployment, and minimal cost. Adding noise to normal data, on the other hand, can confuse a defender or a legal user.

On the other hand, most data obfuscation research focuses on developing a strategy for hiding real information rather than detecting an attacker [25]. SA (Sensitivity analysis) examines how perturbed instances of the method’s input affect the outcome for any particular methodology. With its random input weights, ELM produces a consistent SA, demonstrating the validity of ELM as a classifier in general and SA in particular [64].

### 3.5. Mixing

Mixing is a concept used in security and privacy techniques to limit likability. Mixing solutions employ exchange systems to avoid direct connectivity among networks [20]. While bait-based fraud can help increase intrusion detection by capturing additional data during the Deception, there is no guarantee of success because the assailants will not be engaged in the bait. Furthermore, if more vital information is utilized as bait to lure attackers more efficiently, the bait itself raises danger when competent attackers might deduce signs of system weaknesses based on the baits they have investigated. As a result, combining real and false information to avoid a major danger, such as semi-bait-based deceit, may be a realistic option [25,60].

### 3.6. Honey-x

Most honey-x tactics (e.g., honey files, honeypots, and honey tokens) are designed to deceive enemies into making suboptimal or weak attack decisions by providing incorrect information. This will necessitate the implementation of additional processes or procedures to ensure that ordinary users or defenders are not misled [25]. Honey-x deception methods related to the employment of various technologies such as honey patches, honeypots and other network assets with advanced monitoring capabilities allow network administrators to decipher details about intruders while masquerading as genuine network assets [20]. Honeypots are legal traps placed in a network to detect or deflect unauthorized access to a system. Honeypots are useful tools in understanding an attacker’s intentions [22]. Honeypots are all tools that pull an attacker into a location where the security team wants them to go to assess their purpose and guide them to do things that might expose them to what they are trying to do [21].

### 3.7. Attacker Engagement

Attacker Engagement entails using feedback to change attacker behavior over time, squandering their efforts while enabling network managers to perform counterintelligence actions [20]. The majority of game-theoretic deception models are static games or single-shot dynamic games. However, some preliminary research has looked into multi-period games. They referred to games with many periods as “dynamic” and referred to these interactions as “attacker engagement” [6]. They used a one-sided randomized game to model the attacker. States correspond to network layers in order from left to right. As a result, the attacker remains unnoticed. Rather than ejecting the attacker, the Defense determines when to engage the attacker and gain information [60]. The authors of the paper [6] provided a list of articles that looked into mimesis. Articles mentioned by them on the left-hand side look at honey-x, whereas articles mentioned by them on the right-hand side focus on attacker engagement. There is no one-to-one correspondence between deception species and games. Two alternative methodologies are used to model honey-x. One method employs signaling games to stress the attacker’s beliefs about whether systems are normal or honeypots. Bayesian Nash games are used in the other strategy. This method is based on resource allocation difficulties, and it results in an overall network design that is best for the Defense. The paper [6] lists three approaches for attacker engagement: multiple-period games, the interaction between games and MDPs (Markov Decision Process), one-sided stochastic games.

## 4. Open Problems in CI-Enabled Defensive Deception

The following Figure 5 illustrates the Open Problems in CI-enabled Defensive Deception.

While we can profit from modeling simple attack processes such as active reconnaissance or security breaches, deploying game-theoretic DD in real-world systems and modeling attacks based on a complicated cyber death chain remains difficult. Research has been sparse and has primarily concentrated on reconnaissance assaults in other fields. Large volumes of traffic flow data can be generated in IoT and SDN systems, which can be leveraged to train machine learning models to identify attackers. However, existing DD techniques for those domains do not include machine learning [25].

New security vulnerabilities to machine learning and deep learning algorithms emerge regularly. Even though many learning frameworks, algorithms, and optimisation mechanisms have been suggested, research into learning models’ security is still in its early stages. As a result, machine learning techniques are vulnerable to various threats; hence a Defensive Deception employed with ML/DL can compromise [29].

Despite attempts to include adversarial samples in training models and improve the resilience of learning algorithms, these solutions are still incapable of solving the frequency of operation. As a result, research on safe deep learning models, such as Bayes, deep networks incorporating prior information, will be particularly intriguing soon [29]. 

Designing safe learning algorithms necessitates balancing security, generalization performance and cost. In general, a higher level of security results in a higher overhead or even a lower prediction accuracy of learning algorithms, which makes their implementation more difficult. Implementation of security strategies with less overhead and cost remains a challenge [29].

The efficiency of machine-learning-based deception strategies is dependent on the availability of information about the attacker, their techniques and their targets. In practice, the Defense lacks access to such information, substantially limiting the training of Machine Learning classifiers and detectors. Furthermore, models such as these are frequently presumed when the attacker behaves properly toward its intended victim. However, the efficiency of deception tactics may not be easily quantified if an attacker chooses to fool a defender to remain stealthy [75,76].

It is easier to set up simple deception tactics than configuring and applying security controls to all information systems’ resources. Deception of vital information systems should be given priority. The methods are not difficult, but they require consideration when choosing from the many available options. Implementing deception on certain resources can be challenging and require much expertise to implement and maintain correctly.

The determination of time period, which is the amount of time after which we enforce the MTD in case of an attack, is in general highly specific and constant. Determining an appropriate time period that could be changed according to the situations remains a difficult task [30].

Conflicting security policies which could occur during the deployment of Defensive Deception techniques must be carefully analyzed, as such conflicts may lead to the loss of genuine user packets or the introduction of new attack vectors. Although some research has attempted to discover security policy conflict in the case of a SDN-managed cloud network, it is not immediately evident how it may be applied to MTDs [30] and other Defensive Deception technologies. 

## 5. Future Directions in Defensive Deception

The following Figure 6 illustrates the Future Directions in CI-enabled Defensive Deception.

### 5.1. Honeypot

We need more parameters to judge the efficiency of honeypot techniques. Honeypot quality has been assessed mostly based on detection rate, even though Honeypot’s primary function is to safeguard assets and identify threats. As a result, better metrics for measuring the responsibilities of both defending assets and detecting threats should be developed. An attacker’s perceived level of uncertainty, the number of missed sensitive resources by intruders, the number of attack pathways discovered using honeypots, or the number of vital network elements assaulted are all examples of metrics. Furthermore, for the Defense to choose an appropriate method based on various variables, such as those of numerous parameters such as targets, the efficiency of running honeypot technologies, or productivity deterioration due to deception installation, must be addressed. Additionally, the attack data should be utilized to analyze intruders and generate honey sources. The only metric that captures ML-based honeypots so far is classification accuracy. Additional metrics for ML-based DD approaches, such as creating misleading traffics and network topologies, should be created. Completely automated Deception is ideal for ML-based Defensive Deception tactics such as producing deceptive traffics and network architectures.

### 5.2. Moving Target Defense

Some of the unexplored areas of MTD that can be further researched are as follows: Most of the MTDs developed today primarily focus on computers and computer networks. Research in integrating MTDs on various surfaces of mobile technologies and networks could be beneficial [30]. Apart from this, we need research to determine reasonable periods. Instead of keeping it constant, the time period must be changed based on the attack model [30].

### 5.3. Other Future Directions

The success of a Defensive Deception strategy should be measured by how successfully it misleads intruders. To judge the degree of deceit, the intruder’s perspective and tactics should depend on its belief in the defender’s actions. On the other hand, current research frequently uses system metrics as a substitute for evaluating the performance of a defender’s deceptive strategy [72,73].

Furthermore, deception tactics should not always be used in conjunction with traditional protection services such as intrusion detection, prevention, and alert systems [74]. This is because particular combinations of deploying tactics with traditional security services, for example, employing honeypots in conjunction with intrusion detection and prevention systems, might generate an inefficient overlapping effect. As a result, we should devise a more systematic technique for using both defense services synergistically to provide cost-effective defense services [74].

When dealing with Defensive Deception technologies, a clear understanding and proper compliance of various legal security policies are difficult. To address this, we can implement a persistent feedback loop that evaluates the security policies following the deployment of MTD countermeasures. This can be accomplished by assuring end-to-end regression and integration testing for numerous network traffic instances. Another option is to simulate policy conflicts that may develop as part of the MTD modeling process. This would help us foresee the policy contradictions in case of an MTD deployment, and we can make changes accordingly [30].

Furthermore, greater research into the implications of adaptable cyber Deception and attacker expertise is required. In the future, there is a need to identify a theory of group utility function to permit groups of agents to make decisions. It will focus on automatic deception packet creation, delivery mechanism development and the resultant deception model. It will also examine if any unique defensive deception tactics or counter-deception techniques exist in the cyber environment.

## 6. Conclusions

The struggle for supremacy is being conducted on both sides as artificial intelligence and machine learning continue to progress at a breakneck pace for good and bad purposes [78]. With so much research and development in these fields, increased computational power, the volume and access to enormous amounts of data, and hyper interconnectivity are the mechanisms via which AI and ML advances benefit. Deception-based defenses are potent weapons that have been proven to work in various domains. Their efficacy is based on the fact that they are programmed to exploit key biases to appear realistic but misleading substitutes to the hidden reality [79,80,81,82,83,84,85,86,87,88,89,90,91,92,93,94,95,96,97,98,99,100,101,102,103,104,105,106,107,108]. As a result, one will require a thorough understanding of both offensive and defensive trickery to implement a perfect Deception strategy.

Such methods give defenders a tactical advantage by learning further about their enemies, limiting secondary information breaches in their systems, and better understanding their attackers. The effects of adaptive Defensive Deception on an automated attacker are compared in this study. This article demonstrated how an autonomic system could manage a Defensive Deception system. As examples, certain procedures and methods are offered to investigate proposals and solutions to specific fraud detection and prevention issues. 

AI/MLS models have already proven to be a benefit and a burden in the cybersecurity field. Consequently, current cybersecurity measures are expected to become obsolete, forcing the creation of new countermeasures. Deception tactics based on machine learning that learn and recognize can be a great tool to automatically deploy and maintain the Deception frameworks. Machine Learning can improve Deception by taking into account various factors such as: We must think about the types of datasets used to construct deception tactics.Good datasets for replicating actual things and evaluating false items are required to create plausible fake objects.ML-based deception tactics should use appropriate metrics to capture their effectiveness and efficiency.

Unlike typical defense methods, Deception entails some risk because it necessitates certain contacts with attackers to confuse or mislead them. It is unavoidable to accept the risk if the purpose of protection necessitates long-term Deception. As mentioned in our future direction, DD should be used with other legacy defensive techniques such as intrusion prevention or detection with proper precations to minimize an overabundance of risk. Moving target defense (MTD) or obfuscation tactics have a similar purpose to Deception in creating attackers’ confusion or doubt. Deception, on the other hand, would produce fake objects or information to deceive an attacker’s cognitive perspective or create a false notion, causing the attacker to pick a sub-optimal or bad attack technique, except for obfuscation or MTD, which modifies configuration settings or data based on the current resources of a system. Hence these properties must be considered when we want to deploy MTD or obfuscation as defence [72,77]. Overall, when we employ CI-enabled techniques such as ML/DL in proper conjunction with Defensive Deception, we can safely protect our resources in a very effective manner. However, at the same time, when the CI techniques are implemented blindly without proper consideration of resources or a final goal, it would not result in decreased protection but would also cause a huge wastage of resources.

## Figures and Tables

**Figure 1 sensors-22-02194-f001:**
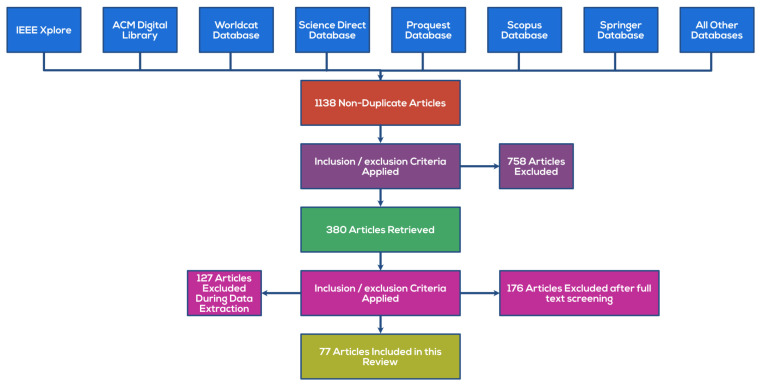
PRISMA flow diagram for the selection process of the research articles used in this review.

**Figure 2 sensors-22-02194-f002:**
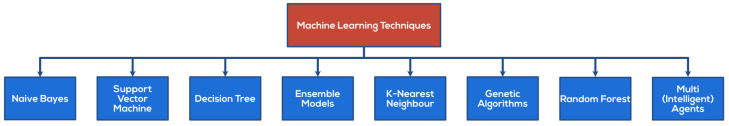
Current machine learning models in defensive deception—nomenclature.

**Figure 3 sensors-22-02194-f003:**
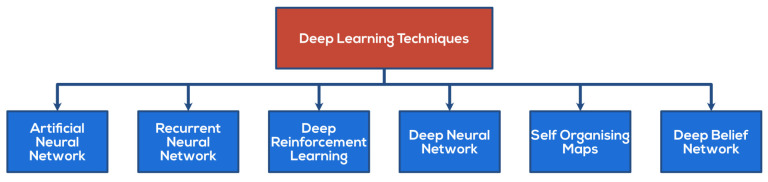
Current deep learning models in defensive deception—nomenclature.

**Figure 4 sensors-22-02194-f004:**
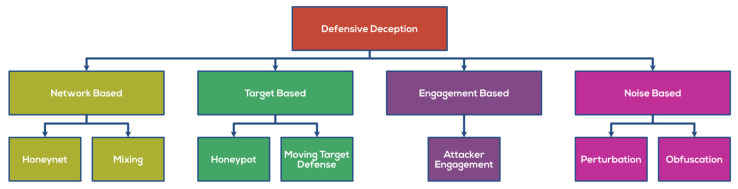
Methods to implement Defensive Deception.

**Figure 5 sensors-22-02194-f005:**
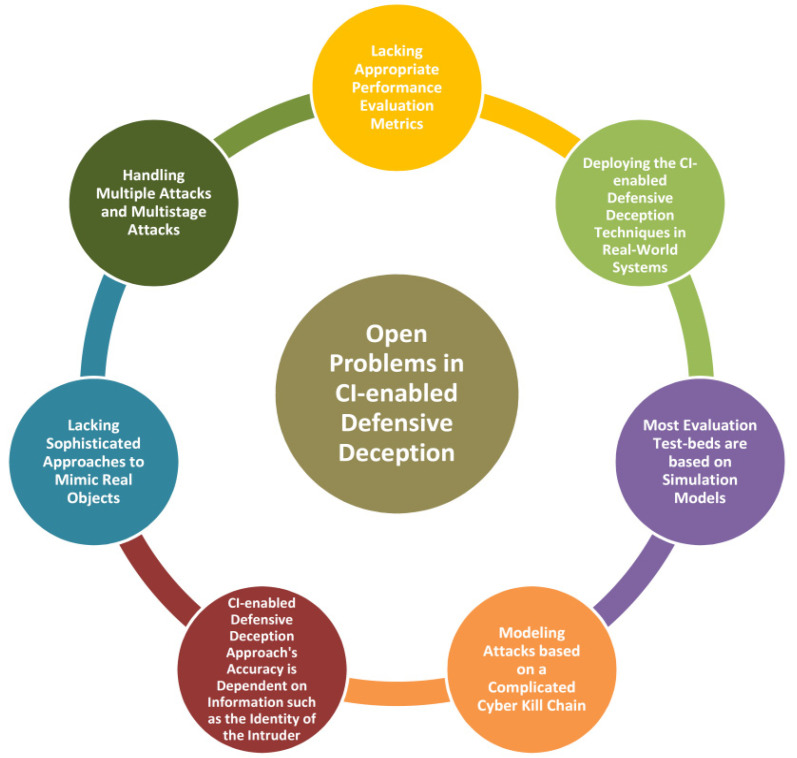
Open Problems in CI-enabled Defensive Deception.

**Figure 6 sensors-22-02194-f006:**
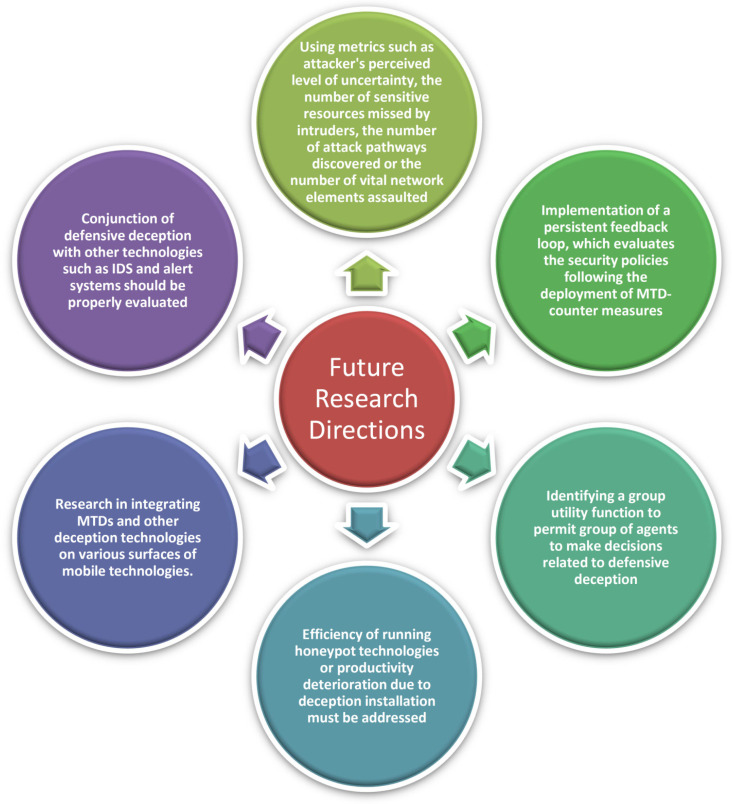
Future directions in Defensive Deception.

**Table 1 sensors-22-02194-t001:** Review articles of the CI-enabled techniques in Defensive Deception (✓: Yes, ×: No).

Ref.	Year	No. of Articles	Brief on Focus (One-Sentence Summary)	CI-Enabled Techniques		Open Challenges	Future Directions
				Machine Learning	Deep Learning		
[39]	2011	28	A Review of Classification Approaches Using Support Vector Machine in Intrusion Detection	✓	×	×	✓
[12]	2012	191	Review article on Nature-Inspired Techniques in the Context of Fraud Detection	✓	×	×	✓
[24]	2012	72	Review article on employment of Data Mining Techniques for financial frauds detection.	✓	×	✓	✓
[18]	2013	62	A review article on Computational Intelligence Models for Insurance Fraud Detection	✓	×	×	✓
[4]	2015	91	A review on application of AI techniques for combatting cybercrime	✓	×	✓	✓
[1]	2018	77	A survey of Artificial Intelligence in Cyber security	✓	✓	×	✓
[16]	2018	41	Review article on employment of machine learning techniques for financial frauds detection.	✓	×	×	✓
[29]	2018	111	A Survey article on Cyber Defensive Techniques employed with the help of Machine Learning algorithms	✓	×	✓	✓
[28]	2019	380	A review of defensive tools and technologies employed in cyberspace	✓	×	✓	✓
[31]	2019	173	A Survey on implementation of adaptive technologies in Moving Target Defense	✓	×	✓	✓
[32]	2020	65	A review article on the implantation of Artificial Intelligence technologies in Electronic Warfare Systems and their applications	✓	✓	×	✓
[34]	2020	145	A Survey article on the implementation of AI, machine learning, and blockchain technology in IoT security	✓	×	✓	✓
[6]	2020	75	A review of deception technologies used in cyber security and user privacy.	✓	×	✓	✓
[26]	2020	83	Review article on AI and machine learning for cybersecurity	✓	✓	×	✓
[30]	2020	175	A Survey article on Moving Target Defenses in order to implement Network Security	✓	×	✓	✓
[25]	2021	187	A Review of Defensive Deception techniques Employed with the help of Game Theory and Machine Learning.	✓	✓	×	✓
Our Review	2022	77	Our review has briefly described various prominent ML and DL models and their use in Deception Technologies.	✓	✓	✓	✓

**Table 2 sensors-22-02194-t002:** A summary of works on machine learning techniques in defensive deception.

Ref.	Deception-Category	Machine Learning Approaches Used	Key Contribution	Limitations
[25]	Honeypots, honey webs, honeynets, honey flies, HMAC, Moving target defense, obfuscation.	K-Means, Support Vector Machine, Hierarchical Grouping, Expectation-Maximization (EM), Bayesian Network (Bayes Net), Decision Tree (DT), Naïve-Bayes Algorithm, C4.5 Algorithm.	This work is primarily concerned with reviewing game-theoretic and machine learning-based Defensive Deception approaches and addressing the findings, limits, and lessons learned from this comprehensive study.	Various deep learning and machine learning approaches such as genetic algorithms, Ensemble Models, Self-organising maps, etc., were not taken into account for Deception.
[30]	Moving target defense	Ensemble model used	This research first classified various Moving Target Defenses according to the surfaces on which these defenses operate. Secondly, they talked about how these MTDs can be put into effect. It discussed the various measures used to assess the effectiveness of MTDs and drew attention towards domains of network security in which the scope of the construction of MTDs is yet to be explored.	The survey did not consider better machine learning and deep learning approaches to implement moving target defenses.
[65]	Honeypot	C4.5, Decision Tree, Naive-Bayes and Bayes Net.	They employed a machine learning method to predict the most vulnerable and easily attackable host in an SDN (Software Defined Networking) network. The security rules for the SDN controller can be developed using the prediction output of machine learning algorithms to prevent unauthorized user access. The experiments revealed that machine learning techniques could enhance security rules for SDN controllers by properly anticipating potential susceptible hosts. The Bayesian Network achieved about 91.68 percent of average prediction accuracy.	New machine learning approaches such as neutrosophic sets were not taken into consideration.
[66]	honeypots	Logistic Regression, SVM, KNN, Naive Bayes, ensemble-based models, Random Forest with Gini, and Extra Tree classifiers with Gini.	They demonstrated that fraudulent clicks on Instagram might boost the popularity index of posts through a variety of tactics with their research. They used honeypots and botnets to launch assaults and collect data from various real and false accounts, such as clicks on various posts. Experimental data show that LR is the most accurate predictor among all the single-based approaches, and among all ensemble-based methods, Random Forest is the best.	They did not consider various other approaches such as hybrid learning models, ANN, etc., in order to validate whether a view is legitimate or fake based on the chosen criteria.
[23]	Obfuscation, Honeypot	Naïve Bayes	They methodically cataloged and ranked the available information system deception options, both offensively and defensively. Then they thought about how Defensive Deceptions could be packaged into “generic explanations” that an attacker would find more persuasive than individual refusals to accept directives.	Latest and better machine learning approaches were not used.
[13]	Obfuscation, Honeypot	Decision Tree	A unique deception strategy was developed for network defenses that achieve reactive unpredictability by combining security postures and probabilistic decision trees. They developed a new grammar for decision-tree that allows analysts to specify and identify potential responses based on warnings, mission processes, security postures, and various asset conditions. A real-time simulation based on an organization and its activities and a historical dataset were used to implement, demonstrate, and assess our technique.	A probabilistic decision system can learn optimal decision tree order execution and security postures. Trees that are manually or automatically generated should potentially be improved to boost speed, especially as they grow larger. Attacks are not learned in the current implementation.
[31]	Moving target defense	Genetic algorithm	They conducted a thorough study of MTD techniques, their core classifications, important design features, frequent attack behaviors addressed by existing MTD implementations. The literature also explored various application fields for the MTD techniques.	This article only briefly investigated the relationship between MTD and other defense systems. There has been little research that looks into the influence of MTD on minimizing attacks after the reconnaissance stage. There has not been much research into the best way to use numerous hybrid MTD approaches. Existing MTD methodologies have limitations in monitoring several parameters of a system’s quality.
[42]	Honeypots	Support vector machine	They described the creation of a novel honeypot-based social bot in order to detect malicious profiles present in social networking groups. Their overall study goal is to look at techniques and propose effective solutions to automatically recognize and filter the profiles of harmful people who target social networking platforms. In order to attract fraudulent accounts, their strategy employs social honeypot personas.	The SVM algorithm used in this article is not suitable for large datasets. It does not perform very well when the dataset has more noise which is the usual case for Twitter accounts.
[61]	Perturbation	Artificial neural network	They demonstrated how ANN might be used to modestly adjust the output probabilities by perturbing the final activation layer of the model. The opponent is forced to ignore the class probabilities, making it necessary to use more queries before successfully performing an attack.	Other machine learning and deep learning approaches were not considered for implementing the system.
[63]	Honeypot	Decision tree	A decision tree is more useful when we have a honeynet rather than just one. Then we may independently test other techniques to determine how well they work and what risks they entail. This is achieved by calculating the average benefit for several honeypots and honeynet layouts, and the one with the highest average benefit is chosen.	Other machine learning algorithms were not used to examine the various scenarios generated by honeynet.

**Table 3 sensors-22-02194-t003:** A summary of works on Deep Learning Models in Defensive Deception.

Ref.	Deception-Category	Deep Learning Models Used	Key Contribution	Limitations
[71]	Money related deception		There is also a new term, Honeyfile, used in this article. Honeyfiles are also used to create confusion and apprehension about the value and location of sensitive data. This method is based on humans’ inability to discern between authentic and bogus information.	There comes a time when cyber security is being scrutinized by the public due to an increasing number of occurrences, even though only a fraction of these instances can be traced back to particular individuals or groups of Blackhats.
[36]	Honeypots, Perturbation	Online Adaptive Metric Learning		Because honeypots are completely “fake systems,” there are a variety of methods available to determine whether the present system is a honeypot or not. They are built with this underlying restriction in mind.
[68]	Honeypots	Recurrent neural network	This study describes a distributed infrastructure capable of deploying decoys across different network segments and managing their physical world perspectives. This solution’s prototype implementation and use case for a boiler model are only two examples of how this new methodology could be used.	To better understand and improve the situation, more research is required. Betterment of fidelity of decoys by generating vendor/product-specific characteristics that include things such as protocols used, ports used, and register point settings.
[53]	Moving target defense, perturbation	Deep neural and deep convolution neural network	They offered MT Deep, a cybersecurity architecture influenced by MTD, as a security service to improve the SAFETY of Deep Neural Network-based classification systems in this study (DNNs). To design the interaction among both MT Deep and users, they used a Bayesian Stackelberg Game. The equilibrium provides the best alternative to the multi-objective problem of lowering misclassified rates on adversarial changed visuals while retaining better classification accuracy on photos images that have not been disturbed.	This article did not examine other neural networks, such as RNN, self-organizing maps, etc.
[15]	Moving target defense	Deep neural network, deep convolution network, and deep reinforcement learning.	The authors have labeled the architecture of RL-based CRM (RL-CRM) according to the types of vulnerabilities it attempts to address. They have shown that the RL-CRM can set up moving target defense, engage attackers for reconnaissance, and lead human attention to mitigate visual weaknesses adaptively and autonomously. Their research revealed that posture-related defense technologies are well-developed, but mitigation options for information-related and human-induced vulnerabilities are still in the early stages of development.	The first hurdle in the learning process is to deal with system and performance limits. Many system limits exist in cyber systems that must be explicitly considered. The improvement of learning speed is a second difficulty. CRM’s (Cyber-Resilient Mechanism) purpose is to restore the cyber system following an attack. Fast learning would allow for a more rapid and resilient response to an attack. Dealing with the non-stationarity of cyber systems is the third difficulty. The environment is assumed to be stationary and ergodic in traditional RL algorithms.
[69]	Honeypot, obfuscation	Deep neural network, deep reinforcement learning	They first introduced SRG (System Risk Graph), a precise adversarial model for extracting specific dangers and internet treatments, such as vulnerabilities in the software and virtualization layers. The adversarial model is updated based on the existing condition system. They proposed a deception rate, which is a statistical parameter for evaluating the efficiency of the deployment method based on SRG. Second, they tweaked a DRL algorithm to develop an adjustable decoy deployment strategy for a rapidly changing internet. Finally, they compared the proposed methodology to existing research using simulations.	This article did not analyze other neural networks such as recurrent neural networks, convolution neural networks, etc.
[70]	Honeypot, obfuscation	Deep neural network, Online Adaptive Metric Learning	A machine learning-based framework for evaluating cyber deception defenses with minimum human participation is developed and implemented. This avoids the problems that come with fraudulent research. Humans, ensuring that automated evaluations are as effective as possible, must be completed prior to human study. Only after this can the next step begin.	They were unable to apply labels to previously unknown categories automatically.
[31]	Moving target defense	Deep neural network, deep convolution network	They conducted a thorough study of MTD techniques, their core classifications, important design features, frequent attack behaviors addressed by current MTD techniques, and implementation found in this article.	This article only briefly investigated the relationship between MTD and other defense systems. There has been little research that looks into the influence of MTD on minimizing attacks after the reconnaissance stage. There has not been much research into the best way to use numerous hybrid MTD approaches. Existing MTD methodologies have limitations in monitoring several parameters of a system’s quality.

**Table 4 sensors-22-02194-t004:** List of various Defense Deception datasets.

Ref.	Year	Authors	Dataset Used	Dataset Size	Format	Details about the Dataset/Brief Description
[115]	2012	Ali Shiravi, Mahbod Tavallaee, Hadi Shiravi, Ali A. Ghorbani,	ISCXIDS2012	16.1 GB	Testbeds from Wireshark	This dataset was developed using a dynamic approach. Their strategy is divided into an Alpha profile and a Beta profile. The Alpha profile uses several multi-stage attack patterns to monitor the anomalous part of the dataset. On the other hand, the Beta traffic generator simulates genuine network traffic, including background noise.
[116]	2013	Gideon Creech, J. Hu	ADFA IDS	5951 records	Training and Validation type	The dataset consists of the password brute force of FTP and SSH. It also includes C100 Webshel payload, Linux Meter-preter, Java-based Meterpreter, and attack vectors with 10 attacks per vector.
[117]	1999	Salvatore J. Stolfo, Wei Fan, Wenke Lee, Andreas Prodromidis, and Philip K. Chan	KDD CUP 1999	2 million connection records with 41 features	relational	It is commonly used as a standard dataset for IDS simulations by researchers.
[118]	2000	Mahbod Tavallaee, Ebrahim Bagheri, Wei Lu, Ali A. Ghorbani	DARPA	5000 records	relational	The 1999 DARPA Intrusion Detection Examination consisted of an off-line and a real-time intrusion detection evaluation.
[119]	2016	Prudhvi Ratna Badri Satya, Kyumin Lee, Dongwon Lee, Thanh Tran, Jason (Jiasheng) Zhang	Likes of Facebook	Records including like are 13,147	relational	A study of fake Facebook Likers obtained from company employees that use the link and honeypot approaches was done. False Likers differed from genuine Likers in terms of liking behaviors, duration, etc.

**Table 5 sensors-22-02194-t005:** Classification of several deception categories.

Reference	Year	Deception Technique	Level of Interaction	Scalability	Resource Level	Goal	Main Attack	Strategy	Domain
[109]	2019	False patch technique	High	Yes	Virtual	Property preservation	Advanced persistent threats	Incorrect facts; Fraud; imitating	Game theory
[110]	2015	Honeypot, designed lure	High	Limited	Virtual	Security for assets; identification of attacks	Probing	Deceiving and imitating	Game theory
[111]	2019	Honeypot	Medium	Yes	Hybrid	Safeguarding assets; identification of attacks	DoS assaults, network drops, and APTs	Deceiving; tries to imitate	IoT
[112]	2018	Honey webs	Low	Yes	Virtual	Preservation of Assets	cyberattack	Deceiving; imitating	Cloud services from the internet
[113]	2018	Deceiving signals	Competitive	NA	Physical	Protection of resources; monitoring of attacks	Advanced persistent threats	Misguiding; concealing; imitating; deceiving	No domain name was provided.
[114]	2021	Misleading Network traffic	Dynamic/high	NA	Physical	Assets preservation	Recon/Investigating	Disguising; mirroring	Cyber–physical system
[42]	2016	Social Honeypot	High	NA	Virtual	Identifying the adversary	The malevolent demeanor of a user	Imitating	A domain is not specified

## Data Availability

Not applicable.

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
