# Peer review of "Leveraging Computational Intelligence Techniques for Defensive Deception: A Review, Recent Advances, Open Problems and Future Directions"

_sensors, 2022, doi:10.3390/s22062194_

Round 1

Reviewer 1 Report

This paper reviews the papers about artificial intelligence and defensive deception such as honeypots, honeytokens, camouflaging, and moving target defense. It is interesting and timely. There are some revisions that need to be done.

  • The references for applications of CI-enabled techniques used in defensive deception in section 2 should be updated. Particularly, sec. 2.2.1 Naïve Bayes [12], section 2.2.2 decision tree [24], section 2.2.5 support vector machine [40][41]. These references are old and newer references should be added. Otherwise, the reader may think these techniques are no longer used today.
  • Zero-day attack should be briefly explained.
  • The full name of MTD should be given.
  • In sec. 1.1 Table 1, references [29][31] both cover all aspects like this manuscript. The relative advantages/novelties should be emphasized.
  • A review paper aims the general readers. The general security measures should be briefly introduced such as physical layer security securities anti-jamming, artificial noise, etc:

Joung, J., Choi, J., Jung, B. C., & Yu, S. (2019). Artificial noise injection and its power loading methods for secure space-time line coded systems. Entropy21(5), 515.

  • Machine learning and deep learning is mutually exclusive (non-overlapping) in this paper. But in most papers, deep learning (using neural network structures such as DNN, CNN, RNN/LSTM, transformer, autoencoder, etc.) is a subset of machine learning such as:

Tseng, S. M., Chen, Y. F., Tsai, C. S., & Tsai, W. D. (2019). Deep-learning-aided cross-layer resource allocation of OFDMA/NOMA video communication systems. IEEE Access7, 157730-157740.

Five Tribes of Machine Learning by Pedro Domingos

https://learning.acm.org/techtalks/machinelearning

The authors should cite these references and emphasize the differences from the other references,

Reviewer 2 Report

TITLE
Leveraging Computational Intelligence Techniques for Defensive Deception:  A Review, Recent Advances, Open Problems and Future Directions

TABLES
Table 2, page 12, ref. [67]: 
"When we have a group of honeypots (a honeynet)"  ==> "When we have a honeynet" 
(because the term 'honeynet' has already been defined). 

Table 3: Ref. [71] was not discussed in text. 
Also: "This method is based on Blackhat’s’ or humans in general’s inability" ==> This method is based on humans' inability
Also: "...as a security service’ " ==> as a security service

Table 4: I was unable to locate some of the authors (Salvatore J. Stolfo, Wei Fan, Wenke Lee, Andreas Prodromidis, and Philip K.
Chan, Joshua Haines) in the refs. 

ENGLISH --- GRAMMATICAL ERRORS, SYNTACTICAL ERRORS
As a general rule, pls avoid starting sentences with pronouns. For instance: 
L372: "He introduced the MTD framework for DNNs". Who do you mean by "He"? 
L475: "They have done quantitative analysis..."
L478: "It also includes a glossary"

The Paper needs proof-reading. There are several grammatical and syntactical errors. For instance: 

Advanced cyber defenses must provide a quick response against attacker activities in real-time scenarios. It demands clever defense systems  ==> 
Advanced cyber defenses must provide a quick response against attacker activities in real-time scenarios. They demand clever defense systems 

L134: "The article [8]..." ==>  Article [8]

L 157-159: 
"Traditional ML and DL vary in the concept that DL methods can be used directly for train and testing the original data 
without having to remove or change its characteristics. [1]"  ==> 
Traditional ML and DL vary in that DL methods can be used directly for training and testing the original data without having to
remove or change their characteristics [1]. 

L176: Pls check this sentence:
"When used effectively, ML’s powerful algorithms create a learning environment for systems, allowing them to accomplish tasks like spotting known and unknown
malicious attacks. [2][3][6]"  ==> 

When used effectively, ML powerful algorithms create a learning environment for systems, 
accomplishing tasks like spotting known unknown malicious attacks [2][3][6]. 

L185: is treated as random variables ==> is treated as random variable 

L374: What is the Stackelberg equilibrium? Pls provide Refs. 

L379:  What are the Boltzmann layers? Pls provide Refs. 

L383: What are the Restricted Boltzmann Machines? Pls provide Refs. 

L403: "While DNN is a powerful complement, allowing it to be used for the cyber security of vast networked systems": Pls check this sentence. (Proposition:  DNN is a powerful complement, allowing it to be used for the cyber security of vast networked systems). 

L422: Self-organizing maps (SOMs) are a popular data visualization tool that allows you to represent a multi-dimensional dataset on a
two-dimensional or three-dimensional map. ==> 
Self-organizing maps (SOMs) are a popular data visualization tool enabling the representation of a multi-dimensional dataset on a
two-dimensional or three-dimensional map.

L479: "Although difficult, the mobility of multiple platforms within a single framework can provide more security benefits than a single
platform movement": Perhaps "Although difficult to implement"? 

L503: honey-X techniques: pls define term here, not in L524. 

L548: "On the left, s look at honey-x, whereas articles look at attacker engagement on the right." <== Pls check this sentence. What do you mean by left and right? 

L552: "Bayesian Nash games":pls define this term or give refs. 

L361: "Deception tactics based on machine learning learn and recognize" ==> Deception tactics based on machine learning which learn and recognize.

TYPOGRAPHICAL ERRORS, PUNCTUATION ERRORS        
As general rules, first pls avoid capitalizing words in the middle of the sentences.  For instance: 
L139: "For Static Decoys condition..."  and 
L199:  other hypotheses.[23] ==> other hypotheses [23].  
etc. 

Second, a reference number in the end of a sentence is placed before the full stop. For instance: 
L195: harm to the system.[12] ==> harm to the system [12]. 

L184, 188: Nave Bayes ==> Naïve Bayes

L248: "Twelve thousand eight hundred eighty-one are malicious (SQL Injections), while 1003 are legitimate." 
Pls use either words or numbers. 

L295: "other techniques such as genetic algorithms or machine learning algorithms are also viable options." ==> 
"genetic algorithms is another viable option." 

L 698: Ieee  ==> IEEE

L354: KDD1999 dataset: is it the “KDD CUP 1999” dataset? 

L376: Sentence should end with full stop. 

L382: "DNN models)." Unmatched parenthesis. 

L511: Sentence should end with full stop.

L531: "Honeypots are legal traps...":  This definition should be given at the beginning, for instance, in L56. 

L586: "Some of the unexplored areas of MTD which can be further researched are - Shifting..." ==> 
Some of the unexplored areas of MTD which can be further researched are: shifting

L612: What is "Unscrewed Vehicles"? Pls explain (define term) or provide refs. 

L619: "leaving the commander unsure which is real" ==> leaving the commander unsure about which is real.

REPETITIONS
Section 1 of this article consists of a brief overview of this article. ==> 
Section 1 consists of a brief overview of this article.  OR Section 1 of this article consists of a brief overview.

L628: "In the cybersecurity field, AI/MLS models have already proven to be a benefit and a burden in the cybersecurity field." ==> 
In the cybersecurity field, AI/MLS models have already proven to be a benefit and a burden. -OR-
AI/MLS models have already proven to be a benefit and a burden in the cybersecurity field.

ACRONYMS 
As a general rule, pls define Acronyms at their first occurrence. For instance: 
RNN (Recurrent Neural Networks) should been defined in line 349, not 361. 
Similarly for DNN (L365) and SDN (L471 & L487). 

Line 79: "CI-enabled techniques"
Pls define "CI".

L277: "the data using support vector machines (SVM)": Acronym SVM has already been defined. 

L286:  Acronym MTD has not been defined. 

L306:  "emerging on their own (MAS)."  Acronym MAS has not been defined. 

L371: "Deep Convolution Neural Networks are three regularly utilized DNN models (DCNN)" ==> 
L371: "Deep Convolution Neural Networks (DCNN) are three regularly utilized DNN models" 

L390:   Acronym PNN has not been defined.     

P.16: is it RLCRM or RL-CRM? Pls choose one. 

P.16:  Acronym SRG has not been defined.     

L471:  Acronym SDN has not been defined.     

L495:  "fake data blended in with real" ==> fake data mixed with real 

L610:  Acronym D&D has not been defined. 

METHODOLOGY
"There are two major promising paths for developing Defensive Deception tactics in this literature. First, attacker and defender
strategies have been commonly described using machine learning..."
--What is the second major promising path? 

"1.1. Contribution of this Survey" 
Pls add a sentence about the contribution of this Survey. 

1.2.1 Search Strategy and Literature Sources:
Various databases like ACM Digital, IEEE, Science Direct, etc."
Pls be specific and precise. 

CONCLUSION
I strongly encourage improving the conclusion. Most part of the Conclusions has an introductory style. The real conclusion is actually the last paragraph.  Pls first recap your work and findings and discuss open problems and future directions. 
What do your results mean to a wider scientific community? 
What are the limitations of this study? (e.g., results written in other languages). 

REFERENCES
Some References are incomplete. For instance: 
[8] Landsborough, J., Carpenter, L., Coronado, B., Fugate, S., Ferguson-Walter, K., & Van Bruggen, D. (2021, January). Towards
Self-Adaptive Cyber Deception for Defense. In HICSS (pp. 1-10)  <== What is HICSS? A conference? 

Round 2

Reviewer 1 Report

No further commentsJ